# Explicit Residence Time Distribution of a Generalised Cascade of Continuous Stirred Tank Reactors for a Description of Short Recirculation Time (Bypassing)

**Peter Toson [1,\*]** [ID] **, Pankaj Doshi [2] and Dalibor Jajcevic [1]**

[1]   Research Center Pharmaceutical Engineering GmbH, Inffeldgasse 13, 8010 Graz, Austria
[2]   Worldwide Research and Development, Pfizer Inc. Groton, CT 06340, USA
\*   Correspondence: peter.toson@rcpe.at; Tel.: +43-316-873-30980

**Abstract:** The tanks-in-series model (TIS) is a popular model to describe the residence time distribution (RTD) of non-ideal continuously stirred tank reactors (CSTRs) with limited back-mixing. In this work, the TIS model was generalised to a cascade of $n$ CSTRs with non-integer non-negative n. The resulting model describes non-ideal back-mixing with $n > 1$. However, the most interesting feature of the n-CSTR model is the ability to describe short recirculation times (bypassing) with $n < 1$ without the need of complex reactor networks. The n-CSTR model is the only model that connects the three fundamental RTDs occurring in reactor modelling by variation of a single shape parameter $n$: The unit impulse at $n \to 0$, the exponential RTD of an ideal CSTR at $n = 1$, and the delayed impulse of an ideal plug flow reactor at $n \to \infty$. The n-CSTR model can be used as a stand-alone model or as part of a reactor network. The bypassing material fraction for the regime $n < 1$ was analysed. Finally, a Fourier analysis of the n-CSTR was performed to predict the ability of a unit operation to filter out upstream fluctuations and to model the response to upstream set point changes.

**Keywords:** residence time distribution; continuous stirred tank reactor; bypassing; Fourier analysis; continuous manufacturing

---

## 1. Introduction

The pharmaceutical industry is currently transforming batch production processes to continuous manufacturing. Continuous manufacturing offers several technical and economic advantages compared to batch processes, such as lower downtimes, better process control, smaller footprints, and ease of scale-up by extending time [1–4]. Better process understanding and control lead ultimately to improved quality of the final product, in a quality-by-design (QdB) framework [3,5,6]. However, material tracking from raw material to finished product remains challenging.

The tool of choice for modelling the flow of material in a continuous process is the prediction and modelling of residence time distributions (RTDs). Each unit operation (e.g., blending, granulation, tableting) is characterised by its RTD. The individual RTDs are then chained together by convolution integrals, in order to calculate the RTD of the overall process. With this information it is possible to predict how long the material remains, on average, in the process (mean residence time, MRT), the response of the system to fluctuations in the material stream (e.g., feeder refills), and to develop process control strategies [7–11].

Residence time distribution modelling is not only used to describe a complete continuous manufacturing line; it is also utilised to describe the complex behaviour of single unit operations within reactor networks. A reactor network contains multiple connected ideal or non-ideal reactors, with a known analytical RTD. The two most common types are the continuous stirred tank reactor (CSTR) and

the plug flow reactor (PFR). These reactor types are too idealised to correctly model the behaviour of a real reactor. However, if these basic models are combined in reactor networks, it is possible to describe the behaviour of a real unit operation, including effects such as dead zones, non-ideal back mixing, and/or bypassing effects. This approach is not limited to pharmaceutical processes, but is also utilised, for example, in chemical reaction engineering and modelling of water treatment processes [12–14].

A well-established model for non-ideal CSTRs is the tanks-in-series (TIS) model, describing a cascade of $n$ CSTRs. This model has an analytical solution for integer $n$, that was first published by MacMullin and Weber [15] and is now part of standard chemical engineering literature [16,17]. A long chain of tanks-in-series results into a sharp RTD peak around the MRT, which is better described by the diffusion model. The diffusion model accounts for axial diffusion in a non-ideal PFR and produces broader peaks in the RTD with increasing diffusion [18–20]. If the fluid velocities inside the PFR are high, compared to the total length of the reactor but still in the laminar flow regime, the RTD is dominated by the parabolic shape of the velocity profile and not by slow diffusion processes. In this case, the RTD is better explained by the convection model [21–24].

Martin [25] generalised the TIS model to non-integer $n \geq 1$ to fine-tune the resulting RTD. This generalisation has been used as a stand-alone model and as a building block in reactor networks [13,24,26–28]. Toson et al. [29] used the model with a narrow parameter range $0.5 < n < 1.1$ to fit the RTD of a continuous powder mixer and linked n to the quality of the mixing process.

The n-CSTR model discussed in this work is an extension of Martin's model [25]. Just like Martin's model, a non-integer value of $n > 1$ allows fine tuning of the RTD shape for varying degrees of limited back-mixing. The n-CSTR model extends the value range to $n < 1$, in order to model bypassing conditions. The n-CSTR model has only one shape parameter, $n$, that can be varied to reach the unit impulse at $n \to 0$, the ideal CSTR at $n = 1$, and the ideal PFR at $n \to \infty$, with the same analytical form. The tanks-in-series model, the diffusion model, and the convection model only connect two of these fundamental RTD shapes (see Figures 1 and 2)

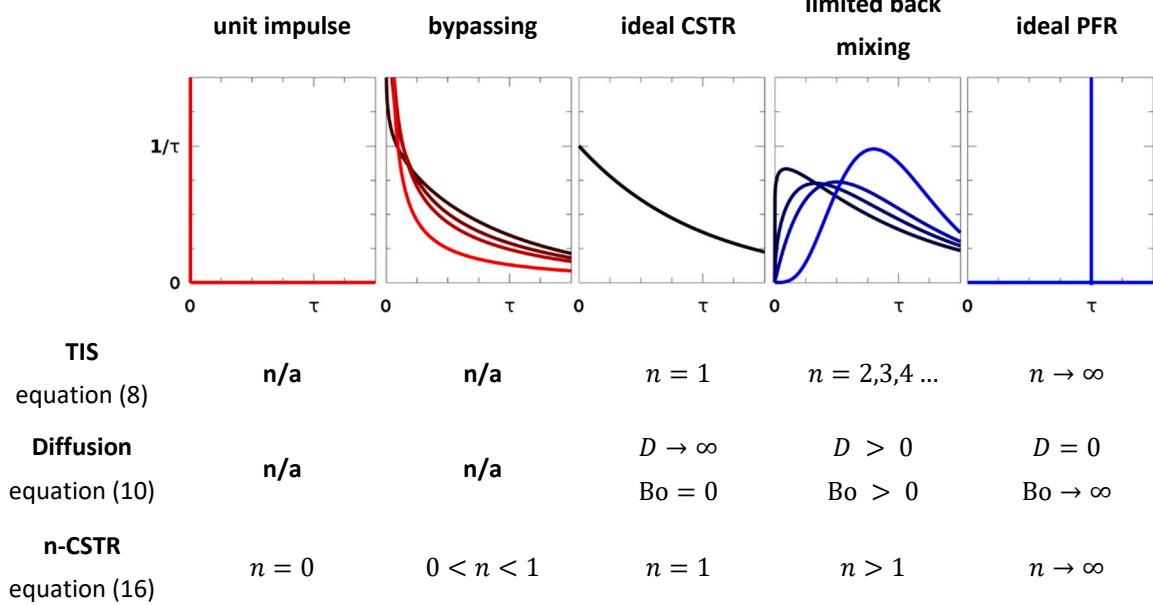

| | unit impulse | bypassing | ideal CSTR | limited back mixing | ideal PFR |
|---|---|---|---|---|---|
| **TIS** equation (8) | n/a | n/a | $n = 1$ | $n = 2,3,4 \dots$ | $n \to \infty$ |
| **Diffusion** equation (10) | n/a | n/a | $D \to \infty$ $\mathrm{Bo} = 0$ | $D > 0$ $\mathrm{Bo} > 0$ | $D = 0$ $\mathrm{Bo} \to \infty$ |
| **n-CSTR** equation (16) | $n = 0$ | $0 < n < 1$ | $n = 1$ | $n > 1$ | $n \to \infty$ |

**Figure 1.** Comparison of the parameter range of tanks-in-series (TIS), diffusion, and generalised cascade of $n$ continuous stirred tank reactor (n-CSTR) models.

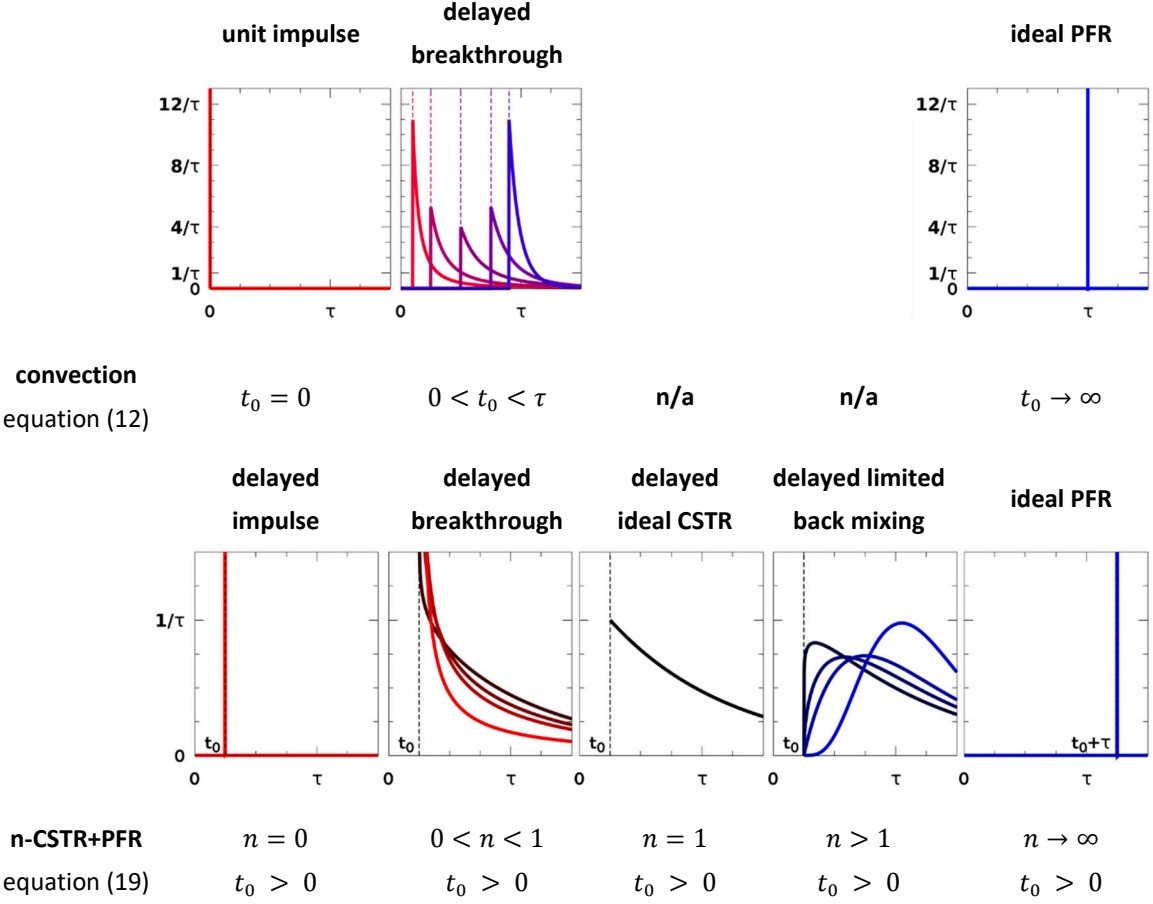

**Figure 2.** Comparison of the parameter range of n-CSTR+PFR (combined n-CSTR + plug flow reactor) model with the convection model.

## 2. Fundamentals of RTD Modelling

The first step in RTD modelling is obtaining the residence time distribution. The experimental setup uses a small amount of tracer material added to the process. The concentration of the tracer material is measured at the relevant outlet. The result is the tracer concentration profile $C(t)$. Once the tracer concentration has decayed completely, the concentration profile is normalised to the so-called $E(t)$ of the RTD:

$$E(t) = \frac{C(t)}{\int_0^\infty C(t) \cdot \mathrm{d}t} \tag{1}$$

The $E(t)$ curve describes the distribution of exit times—its peaks indicate the time where most of the tracer material is discharged. As $E(t)$ is a normalised distribution with integral 1, it is possible to calculate statistical indicators (mean residence time $\tau$, standard deviation $\sigma$) directly from the $E(t)$ curve:

$$
\begin{aligned}
\tau &= \int_0^\infty t \cdot E(t) \cdot \mathrm{d}t \\
\sigma^2 &= \int_0^\infty (t - \tau)^2 \cdot E(t) \cdot \mathrm{d}t
\end{aligned}
\tag{2}
$$

The key to RTD modelling is the ability to combine multiple RTDs—multiple $E(t)$ curves—into one process-level RTD. If a process with an RTD $E_1(t)$ feeds into a process with a different RTD $E_2(t)$, the combined RTD is calculated by the convolution integral

$$
\begin{aligned}
\mathrm{RTD}(t) &= \int_{\theta=0}^{\theta=t} E_1(t - \theta) \cdot E_2(\theta) \cdot \mathrm{d}\theta \\
&= \int_{\theta=0}^{\theta=t} E_1(\theta) \cdot E_2(t - \theta) \cdot \mathrm{d}\theta
\end{aligned}
\tag{3}
$$

A shorthand notation for convolution uses the * operator. The convolution is symmetric:

$$\text{RTD}(t) = (E_1 * E_2)(t) = (E_2 * E_1)(t) \tag{4}$$

The convolution operation has a neutral element—the Dirac (or unit) impulse $\delta(t)$. If any distribution $E(t)$ is convolved with the Dirac impulse, the result is the distribution itself, as shown in Equation (5). Tracer experiments are the practical application of this equation: if a tracer impulse is added at the inlet, the RTD of the process can be observed as a tracer concentration profile at the outlet. Figure 1 shows the unit impulse in red.

$$(E * \delta)(t) = E(t) \tag{5}$$

## 3. RTD Models and Their Limits

### 3.1. Ideal Plug Flow Reactor (PFR)

The ideal continuous stirred tank reactor (CSTR) and the ideal plug flow reactor (PFR) are the most basic models for continuous reactors and are widely used model prototypes in the chemical engineering community [16,17]. Both are characterised by their mean residence time $\tau$. If a tracer pulse is added at the inlet of the ideal PFR, the response at the outlet is a delayed pulse of the same shape. Therefore, the residence time distribution of the ideal PFR is a Dirac impulse delayed by $\tau$—see Equation (6). The RTD of the ideal PFR is shown in Figure 1 in blue.

$$\text{RTD}_{\text{PFR},\tau}(t) = \delta(t - \tau) \tag{6}$$

### 3.2. Ideal Continuous Stirred Tank Reactor (CSTR)

The ideal CSTR is characterised by perfect back mixing. Each portion of the material has the same chance to be discharged at the outlet, regardless of how long it has already been inside the CSTR. If some tracer is added to the CSTR, it is perfectly mixed within the tank and a portion of the tracer material is immediately visible at the outlet. The constant discharge chance leads to an exponentially decaying tracer profile. The assumption of perfect mixing directly leads to the typical exponential RTD shape of the ideal CSTR (black line in Figure 1). The RTD of an ideal CSTR with mean residence time $\tau$ is given by:

$$\text{RTD}_{\text{CSTR},\tau}(t) = \frac{1}{\tau} \exp\left\{-\frac{t}{\tau}\right\} \tag{7}$$

### 3.3. Tanks-in-Series (TIS)

A popular model for non-ideal CSTRs is the cascaded CSTR or tanks-in-series (TIS) model. A defined number of CSTRs ($n$) are chained together such that the inlet of one CSTR is connected to the inlet of the next. Due to the fact that material can only flow from one CSTR in the cascade to the next, but not backwards, the TIS model describes imperfect back mixing. Each CSTR in the chain has the same mean residence time of $\tau/n$. The RTD of the TIS model can be calculated by convolving the RTD of the CSTRs n times with itself. MacMullin and Weber [15] were the first to derive a general formula for the TIS model with $n$ CSTRs in series:

$$\text{RTD}_{n,\tau}(t) = \frac{t^{n-1}}{(n-1)!} \left(\frac{n}{\tau}\right)^n \exp\left\{-\frac{t\,n}{\tau}\right\} \tag{8}$$

For $n = 1$, Equation (8) reduces to Equation (7), the RTD of an ideal CSTR. With higher values of n, the peak of the RTD moves from $t = 0$ closer to $t = \tau$, and the height of the peak increases.

The theoretical limit for $n \to \infty$ is a singular peak at $t = \tau$, which is the RTD of an ideal PFR with the same mean residence time $\tau$:

$$\mathrm{RTD}_{n\to\infty,\tau}(t) \to \delta(t - \tau) = \mathrm{RTD}_{\mathrm{PFR},\tau}(t) \tag{9}$$

Figure 1 shows the RTD shapes and transition from ideal CSTR, over imperfect back-mixing, to PFR as a black–blue gradient.

### 3.4. Diffusion Model

A classical model for non-ideal plug-flow reactors is the dispersion model. The basis for this is the dimensionless partial differential equation:

$$\frac{\partial C}{\partial \theta} = \left(\frac{D}{uL}\right) \cdot \frac{\partial^2 C}{\partial x^2} - \frac{\partial C}{\partial x} \tag{10}$$

with the concentration $C = C(x, \theta)$ as a function of non-dimensional position in the tube $x$ and non-dimensional time $\theta = t/\tau$, the axial diffusion coefficient $D$, the velocity of the material along the tube $u$, and the length of the tube $L$.

The residence time distribution is then calculated from the concentration over time at the end of the tube ($x = 1$). The exact analytical solution for closed boundary conditions is unknown. There are approximations for low diffusion $(D/uL) < 0.01$, where the RTD is a Gaussian distribution centred at the mean residence time $\tau$ and variance $\sigma^2 = (D/uL)$. For higher levels of diffusion, the results have to be obtained numerically [16,18,20]. The results show that as the diffusion in the non-ideal increases, the RTD assumes more and more the shape of the TIS model with low values of $n$. As $D \to \infty$, the solutions approach the exponential distribution of the ideal CSTR.

In a sense, the CSTR and PFR reactor models are mirror images: the worst possible CSTR imaginable with no back mixing at all ($n \to \infty$) is the ideal PFR ($D = 0$), and conversely, the worst possible PFR with infinitely high diffusion ($D \to \infty$) has the perfect mixing properties of a CSTR ($n = 1$). The limits and RTD shapes are summarised in Figure 1.

### 3.5. Convection Model

For very viscous fluids or very short tubular reactors, the dispersion model for non-ideal plug-flow reactors may not feasible. In this case, the RTD is a consequence of the characteristic parabolic velocity profile of a laminar flow, not driven by diffusion. Ananthakrishnan et al. [21] derived an RTD for a pure convective flow with mean residence time $\tau$.

$$\mathrm{RTD}_{\mathrm{conv},\tau}(t) = \frac{\tau^3}{2 \cdot t^3} \quad \text{if} \quad t \geq \frac{\tau}{2} \tag{11}$$

If the convection model has a dead time $t_0 = \tau/2$ before the first material exists the tube, followed by a $t^{-3}$ decay. This effect is independent from the velocity of the flow. There are some generalisations to this model (for example, References [30,31]), but Gutierrez et al. [24] gave a dimensionless generalisation which is parametrised with the normalised breakthrough time $\theta_0 = t_0/\tau$ with $0 \leq \theta_0 \leq 1$:

$$\mathrm{RTD}_{\mathrm{conv},\theta_0}(\theta) = \frac{1}{1-\theta_0} \cdot \frac{1}{\theta} \cdot \left(\frac{\theta_0}{\theta}\right)^{\frac{1}{1-\theta_0}} \quad \text{if} \quad \theta \geq \theta_0 \tag{12}$$

The $\theta_0 = 0.5$ Equation (12) simplifies to a non-dimensional version of Equation (11). Equation (12) can also be rewritten to a dimensional version with mean residence time $\tau$ and breakthrough time $t_0$ with $0 \leq t_0 \leq \tau$:

$$\mathrm{RTD}_{\mathrm{conv},\tau,t_0}(t) = \frac{\tau^2}{t \cdot (\tau - t_0)} \cdot \left(\frac{t_0}{t}\right)^{\frac{\tau}{\tau - t_0}} \quad \text{if} \quad t \geq t_0 \tag{13}$$

An interesting feature of this parameterisation is that the two limiting cases are the unit impulse $\delta(t)$ as the breakthrough time approaches zero ($t_0 \to 0$) and a delayed Dirac impulse $\delta(t - \tau)$—the RTD of

an ideal PFR—as the breakthrough time approaches the mean residence time ($t_0 \to \tau$). The RTD shapes and limits of the convection model are summarised in Figure 2, the transition from unit impulse to PFR is shown as a red–blue colour gradient.

## 4. Generalised Cascade of *n* Continuous Stirred Tank Reactors: The n-CSTR Model

The diffusion, TIS, and convection models cover a wide variety of RTD shapes. However, because of their different analytical forms, the only way to describe transitions between a bypassing condition and a non-ideal mixing condition, is to build reactor networks containing convective elements and CSTR cascades or utilising parallel CSTRs in reactor networks [13,25,28,32]. The TIS model has been generalised to a non-integer number of CSTRs in series [25], but it is also possible to utilise less than one CSTR to describe short-circuiting and bypassing effects [29]. The result is the generalised n-CSTR model.

### 4.1. The Γ(n) Function

The basis for the generalised n-CSTR Model is the TIS model in Equation (8) which is limited to natural numbers of *n*. This limitation comes from the factorial expression $(n-1)!$ in Equation (8). If the factorial could be replaced by a real-valued function, it would be possible to provide an analytical residence time distribution of n CSTRs for *any* value of *n*. Such a function exists—the gamma function $\Gamma(n)$.

The gamma function traces back to Euler, who defined the first analytic continuation of the factorial [33]. The gamma function is defined with an absolute converging infinite integral:

$$\Gamma(n) = \int_0^\infty x^{n-1} \cdot e^{-n} \cdot dx \tag{14}$$

and suffices the following conditions:

$$\begin{aligned}
&\Gamma(0) = \Gamma(1) = 1 \\
&n \cdot \Gamma(n) = \Gamma(n+1) \; \forall \, n \in \mathbb{R} \\
&\Gamma(n) = (n-1)! \; \forall \, n \in \mathbb{N}
\end{aligned} \tag{15}$$

The gamma function is available in many programming languages and tools, for example C [34], Python via the SciPy library [35], Matlab [36], and Microsoft Excel [37]. With the help of the gamma function, it is possible to re-write Equation (8) to be defined for any positive *n*:

$$\mathrm{RTD}_{n,\tau}(t) = \frac{t^{n-1}}{\Gamma(n)} \cdot \left(\frac{n}{\tau}\right)^n \cdot \exp\left\{-\frac{t\,n}{\tau}\right\} \tag{16}$$

### 4.2. Influence of Shape Parameter n

The RTD of a generalised cascade in Equation (16) has two parameters: the mean residence time $\tau$ and the number of CSTRs in the cascade *n*. The generalisation allows a finer tuning of the shape of the RTD, because non-integer values of *n* are allowed. While a cascade of 1.5 CSTRs may seem counter-intuitive at first, non-integer values can occur when analysing experimental RTD curves. The textbook example is calculating *n* from the mean residence time and variance of a measured RTD:

$$n = \frac{\tau^2}{\sigma^2} \tag{17}$$

With the limitation to integer n in the standard TIS model, it is usually recommended to round *n* to the nearest integer [17,38]. With the generalised n-CSTR model it is possible to use the non-integer *n* directly, to fine-tune the shape of the RTD (Figure 3).

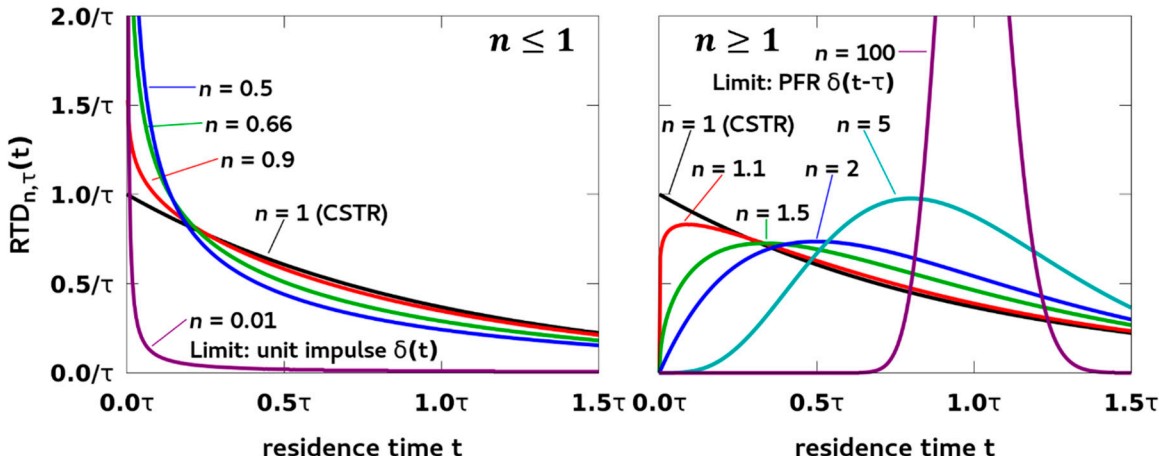

**Figure 3.** Influence of parameter $n$ on the RTD shape for $n \leq 1$ and $n \geq 1$.

The number of $n$ CSTRs is also connected to the Bodenstein number Bo, which is indirectly proportional to the diffusion in the system. Thus, calculating the Bodenstein number for a process and relating it back to the number of CSTRs in the cascade can result into a non-integer $n$. The relation between n and the Bodenstein number Bo is given by Elgeti [39]:

$$n = \frac{\text{Bo}}{2} + 1 \tag{18}$$

As a side note, the Bodenstein number Bo and the Peclet number Pe are sometimes used interchangeably in this context, although the numbers have a slightly different meaning [16]. Just like the TIS model, the generalised n-CSTR cascade connects the ideal CSTR model ($n = 1$, Bo $= 0$, $D \to \infty$) and the ideal PFR model ($n \to \infty$, Bo $\to \infty$, $D = 0$) with a region of limited back mixing ($1 < n < \infty$).

The novelty of the $n$-CSTR model is expansion to the case $n < 1$. The resulting RTDs show a sharp peak at $t = 0$, while still maintaining the exponentially shaped tail. The initial peak describes a short-circuiting or bypassing behaviour without changing the overall mean residence time $\tau$. This behaviour is impossible to describe with the classical TIS and diffusion models. A common practice to describe bypassing is building complex reactor networks utilising TIS or diffusion models with short residence times, e.g., [13,25].

A way to build intuition for a cascade with less than one CSTR is to consider a single CSTR, but it is not fully utilised and thus some material is able to bypass the mixing, moving directly from inlet to outlet. It makes sense to describe this scenario with $n < 1$. A real-world example is a vertical continuous mixing device described in Reference [29]. The construction is based on an ideal CSTR and for a range of operating conditions it behaves exactly like one ($n = 1$). Small deviations from these operating conditions cause small deviations from the ideal CSTR behaviour which fall either on the limited back-mixing side ($n > 1$) or on the increased initial peak side ($n < 1$). As it is possible to describe both non-ideal cases with the n-CSTR model with only one parameter, fitting the shape parameter $n$ to the obtained RTDs provides a way to describe the quality of the process with a single number.

As n becomes smaller and smaller, the RTD peak becomes sharper and converges to the unit impulse $\delta(t)$ for $n \to 0$ (Figure 3). Intuitively, if there are no CSTRs in the cascade at all, there is nothing that could change the RTD of any incoming material. A visual comparison of the generalised n-CSTR model with the TIS and diffusion model is given in Figure 1.

The behaviour for $n < 1$ is similar to the convection model; however, the convection model adds a delay time $t_0$, whereas the peak stays at $t = 0$ in the generalised CSTR cascade. If needed, the n-CSTR model can be combined with an ideal PFR to add a delay time:

$$
\begin{aligned}
\text{RTD}_{n,\tau,t_0}(t) &= \big(\text{RTD}_{\text{PFR},t_0} * \text{RTD}_{n,\tau}\big)(t) \\
&= \text{RTD}_{n,\tau}(t - t_0) \quad \text{if} \quad t \geq t_0
\end{aligned}
\tag{19}
$$

The mean residence time of this system is $\tau + t_0$. The parameter range and possible RTD shapes of this model is summarised and compared to the convection model in Figure 2.

### 4.3. Quantification of Bypassing Material Fraction

Any n-CSTR values in the range $0 < n < 1$ are considered to model bypassing of material. The fraction of material which bypasses the reactor/mixer increases for smaller values of $n$. In order to quantify the bypassing material fraction, the cumulative residence time distribution has been calculated by numerically integrating over Equation (16), up until a certain bypassing time threshold $t_{\max}$.

Figure 4 shows the bypassing material fractions for $t_{\max} = 0.01\,\tau$, $0.05\,\tau$, $0.1\,\tau$. By defining the bypassing fraction as the integral of the first part of the RTD, the ideal CSTR at $n = 1$ also shows a small bypassing fraction. This fraction corresponds to the small amount of the newly added material which is perfectly mixed and immediately visible at the outlet. The bypassing fraction in the initial peak of the RTD increases slowly for n values close to 1 and shows a steep increase for n values close to 0. The limiting case for $n \to 0$ is the Dirac impulse, where all material instantly leaves at $t = 0$.

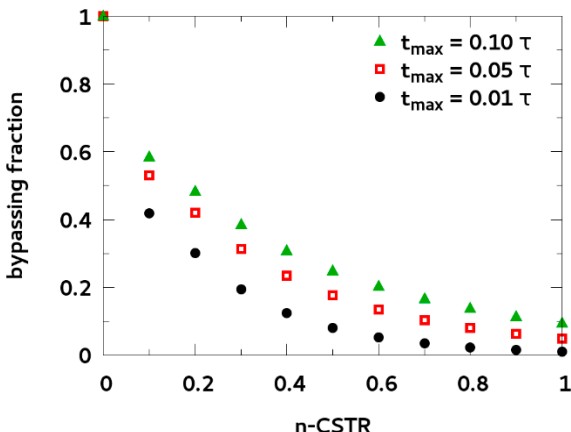

**Figure 4.** Bypassing mass fraction in the first 1%, 5%, and 10% of the mean residence time $\tau$ as function of the number of CSTRs $n$.

### 4.4. Filtering of Mass Flow Fluctuations in a Continuous Manufacturing Line

The RTD of the generalised n-CSTR model can in principle be used anywhere the models of the non-ideal CSTR or non-ideal PFR have been applied: description of complete processes, as a model for a single unit operation, or as a building block in a reactor network.

The idea of RTD modelling is to characterise each unit operation in the continuous manufacturing process (e.g., mixing, granulation, tableting) with the residence time distribution. The goal is to predict how the fluctuations in the initial material stream propagate throughout the process and effect the final product quality. If the RTD—in our case the generalised n-CSTR model $\text{RTD}_{n,\tau}(t)$—and the input mass flow $m_{in}(t)$ is known, it is possible to calculate the mass flow at the outlet $m_{\text{out}}(t)$ with a convolution integral [8]:

$$
\dot{m}_{out}(t) = \big(\dot{m}_{in} * \text{RTD}_{n,\tau}\big)(t)
\tag{20}
$$

Vertical continuous mixing devices have broad RTDs that filter out mass fluctuations in the input stream. Typical mean residence times in a continuous mixing device are in the range of 100 s, and good mixing behaviour is indicated with number of CSTRs *n* close to 1 [29].

Even if no data for $m_{in}(t)$ are available, it is possible to characterise the filterability of the RTD with Fourier analysis (for details see Reference [40]). The filterability Fe(f) indicates how much the frequency f is damped by the RTD. The filterability spectrum has been calculated using Matlab's built-in function fft() for calculating the fast Fourier transformation. The input is the RTD curve given by the n-CSTR model, with τ = 100 s in Equation (16). The RTD curve has been sampled with a constant time step Δ*t* = 0.01 s up to 6000 s, thus the frequency spectrum is available up to 50 Hz with a resolution of 1/3000 Hz.

The filterability spectra are shown in Figure 5. All spectra start at one for 0 Hz, because it is impossible to filter a constant input stream. As the frequency increases, the amplitude of the fluctuations is reduced. For example, an RTD with *n* = 1 and τ = 100 s damps frequencies f = 1 Hz to Fe(f) ≈ 1/1000 of its initial value (see Figure 5a).

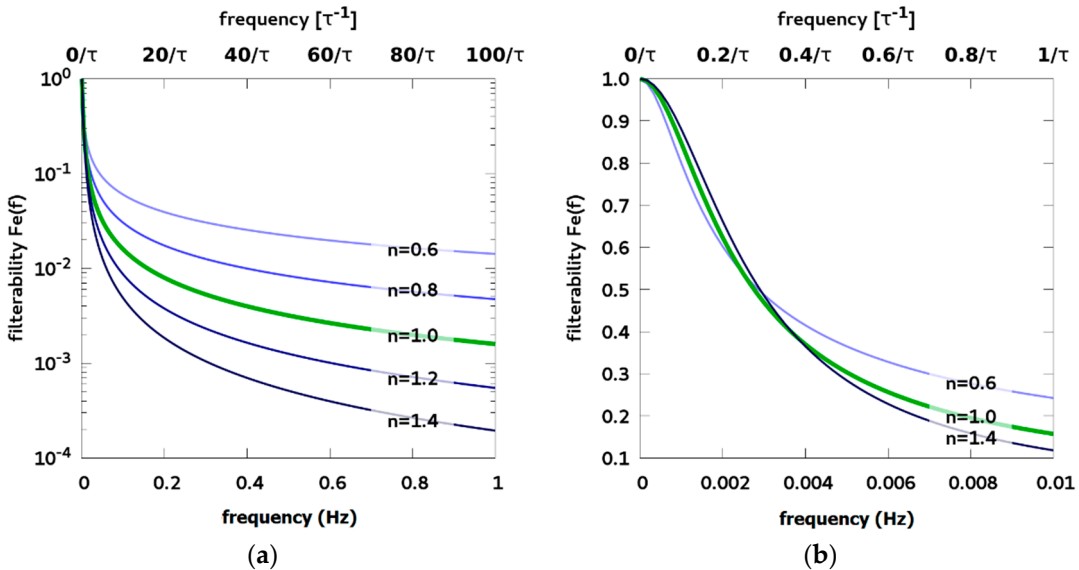

**Figure 5.** Filterability of a continuous mixing device with τ = 100 s and n-CSTR close to 1. (**a**) Higher values of n indicate better filtering of higher frequencies. (**b**) Very low frequencies below $0.3\,\tau^{-1}$ (i.e., drifts in the data that are longer than one third of the mean residence time τ) are better filtered with lower values of n-CSTR.

Generally speaking, residence time distributions with higher mean residence times are better frequency filters. The filterability spectrum is inversely proportional to the mean residence time τ. Increasing τ effectively compresses the filterability spectrum, reaching the low frequency amplitudes faster (see normalised frequencies $\tau^{-1}$ in Figure 5).

The value of *n* changes the shape of the frequency filter: Lower values of *n* filter very low frequencies $f < 0.3\tau^{-1}$ slightly better (Figure 5b), but they also show significantly worse damping of higher frequencies (see Figure 5a). This can also be explained with the bypassing behaviour described by the n-CSTR model with *n* < 1: Rapid changes (high frequencies) in the inlet material stream are immediately visible at the outlet due to bypassing. However, slow drifts in the inlet stream (low frequency) are damped by the narrow and long tail of the remaining RTD after the initial peak.

Another way to analyse the filterability of the n-CSTR model is to construct an inlet condition and perform the convolution integral in Equations (3) and (20). Figure 6a shows the response of the n-CSTR model to a rectangular inlet condition with a length of one MRT. While appearing artificial, this rectangular inlet condition is a good model for set point changes occurring in a continuous manufacturing process [9,10,29]. Even though the change in the inlet condition (e.g., mass flow,

concentration) persists for one mean residence time, the peak at the outlet is damped significantly compared to the height of the rectangle.

Figure 6b plots the outlet peak for different rectangle widths Δt and n-CSTR values close to 1. Higher n-CSTR values lead to better damping of the rectangular input and lower peaks at the outlet. This effect is more visible at narrower rectangle widths: the longer the rectangular inlet condition, the harder it is to dampen.

Figure 5b shows that lower n-CSTR values lead to a better damping behaviour at lower frequencies and higher durations, which seem to contradict the results from Figure 6b. The reason is that even rectangles with a high duration (and thus low frequency) have a complex frequency spectrum with high frequencies occurring at the flanks of the rectangle. Although lower n-CSTR values damp very low frequencies better, the filterability of high frequencies is significantly worse. The result is an overall higher response at low n-CSTR values, as shown in Figure 6b.

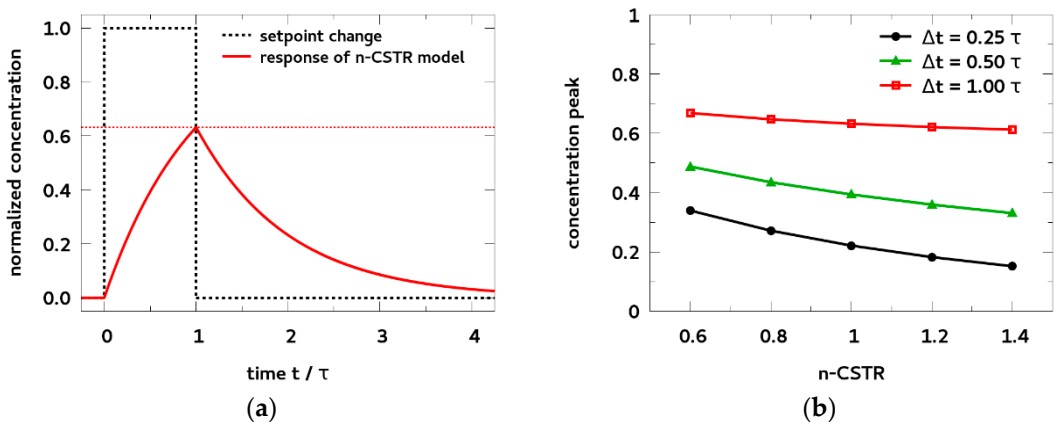

**Figure 6.** (**a**) Response of the n-CSTR model to a rectangle with a length $\Delta t = \tau$ and $n = 1$. (**b**) Concentration peak at the outlet for n-CSTR values around 1 for 3 different rectangle lengths $\Delta t = 0.25\,\tau, 0.5\,\tau, 1.0\,\tau$.

## 5. Conclusions

In this work, the tanks-in-series (TIS) model has been generalised to a cascade of an arbitrary number of continuous stirred tank reactors (n-CSTR model). Several unique properties of the n-CSTR model have been discussed.

The n-CSTR model does not only allow fine-tuning of the well-known TIS model with $n > 1$, but it also expands to a new class of residence time distributions describing short-circuiting and bypassing effects with $n < 1$. The already established convection model offers similarly shaped residence time distributions with a high initial peak for bypassing effects; however, the parameterisation with $t_0$ changes both shape and position of the peak simultaneously. Changing the number of CSTRs in the n-CSTR model changes only the shape of the distribution, but the start of the RTD always remains at $t = 0$. Thus, if desired, it is trivial to choose the starting position of the RTD of the n-CSTR model by applying an additional offset ($t_0$).

The n-CSTR model is the only model which connects the unit impulse ($n \rightarrow 0$), a bypassing regime ($0 < n < 1$), the ideal CSTR ($n = 1$), a limited back-mixing regime ($n > 1$), and the ideal plug flow reactor ($n \rightarrow \infty$) with the same analytical form for any given mean residence time $\tau$, by only adjusting one shape parameter: $n$. Bypassing effects in processes with near ideal CSTR behaviour can be modelled with shape parameters $n < 1$, without the need for reactor networks with multiple fitting parameters. The smooth transition between bypassing and limited back-mixing enables a simple curve fitting with the shape parameter $n$. This parameter seconds as a descriptor for the mixing quality of a process, with values close to 1 being optimal (ideal CSTR).

Lastly, the applicability of the generalised n-CSTR model has been demonstrated by analysing bypassing fractions, dampening behaviour of fluctuations occurring in a continuous manufacturing line, and the response to set point changes.

**Author Contributions:** Conceptualisation: P.T., P.D., and D.J.; Methodology: P.T. and D.J.; Project administration: P.D.; Supervision: D.J.; Visualisation: P.T.; Writing—original draft: P.T.; Writing—review and editing: P.D. and D.J.

**Funding:** This research received no external funding.

**Acknowledgments:** The Research Center Pharmaceutical Engineering was funded by the Austrian COMET Program under the auspices of the Austrian Federal Ministry for Transport, Innovation and Technology (BMVIT), the Austrian Federal Ministry of Digital and Economic Affairs (BMDW), and by the Federal State of Styria (Styrian Funding Agency SFG). COMET is managed by the Austrian Research Promotion Agency FFG.

**Conflicts of Interest:** The authors declare no conflict of interest.

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
