# Peer review of "Explicit Residence Time Distribution of a Generalised Cascade of Continuous Stirred Tank Reactors for a Description of Short Recirculation Time (Bypassing)"

_processes, doi:10.3390/pr7090615_

Round 1

Reviewer 1 Report

Toson et al present the so-called n-CSTR model, i.e. replacing the (n-1)! term of the conventional TIS by the gamma function. Consequently, non-integer n values are allowed. As such the model is decoupled from the original idea of describing RTD n tanks in series only. Instead, non-ideal tanks with n<>1 are considered instead.

While the reviewer agrees with the implementation of the authors' idea, his basic concern is about novelty and significance of the approach. As correctly stated by the authors, non-integer values of n are likely to be derived from real experiments. Hence, their use with respect to TIS is not new, but fairly common. The novelty of the authors' approach lies in the application of non-integer n-values ab initio. Indeed, the gamma function has not been applied so far, but what is the purpose? Is there an unequivocal way for setting proper non-integer n values prior to any experiments e.g. considering by-passes? This is, what the authors have in mind. Or shouldn't n setting rather be based on experimental retrospective analysis, which is the common method now? Frankly speaking, the reviewer supports the latter and qualifies the first as a niche approach.

Author Response

Answer to reviewer 1

We thank the reviewer for the discussion. Based on the suggestions, we clarified the novelty of the n-CSTR model: The ability to model bypassing effects without building reactor networks. The application we have in mind is indeed a retrospective analysis of RTD data, in our case of a continuous mixing device (see ref [29]). The manuscript now explicitly uses ref [29] as an application example and connects the n-CSTR value to the mixing quality in the discussion.

[29] Toson, P.; Siegmann, E.; Trogrlic, M.; Kureck, H.; Khinast, J.; Jajcevic, D.; Doshi, P.; Blackwood, D.; Bonnassieux, A.; Daugherity, P.D.; et al. Detailed modeling and process design of an advanced continuous powder mixer. Int. J. Pharm. 2018, 552, 288–300. https://doi.org/10.1016/j.ijpharm.2018.09.032

Reviewer 2 Report

This manuscript focuses on the generalization of the well known tanks-in-series (TIS) model, allowing to successfully describe short-circuiting and bypassing effects.

The generalized model presented could be of considerable interest for chemical engineers, and this work could contribute to expanding the knowledge on this topic. The introduction and contextualization of the subject are very well written; the fundamentals of reactor modelling are clearly and comprehensively presented.

However, the manuscript would need significant improvements before being accepted for publication in Processes. Both the structure and style of the manuscript appear to be more suitable for a text book than for a journal. The actual contribution of the authors' work should be more clearly specified, and differentiated from the theory previously described in the literature. Also, the abstract should be similarly improved. Filterability and response analysis of the model should be expanded, and explained in more detail.

Finally, English language should be revised; there are a number of spelling and grammar mistakes.

Author Response

Answer to reviewer 2

We thank the reviewer for the discussion and critique. We emphasized the novelty of the n-CSTR model, especially the region with n<1 to model bypassing effects. The result is a unique model offers a smooth transition from unit impulse over bypassing, ideal CSTR, and limited back-mixing regimes to the PFR. The "textbook" introduction is here to build the theoretical basis and to show the limits ("limits" being both mathematical limits and limitations) of existing models. This theoretical foundation is the novelty of the manuscript, the authors have already used the n-CSTR model in a more hand-wavy way in ref. [29], where the focus was the continuous mixing process.

Based on the suggestions, we have expanded the discussion on the filterability and response section and made minor language revisions throughout the manuscript.

[29] Toson, P.; Siegmann, E.; Trogrlic, M.; Kureck, H.; Khinast, J.; Jajcevic, D.; Doshi, P.; Blackwood, D.; Bonnassieux, A.; Daugherity, P.D.; et al. Detailed modeling and process design of an advanced continuous powder mixer. Int. J. Pharm. 2018, 552, 288–300. https://doi.org/10.1016/j.ijpharm.2018.09.032

Reviewer 3 Report

This manuscript reports an extended version of generalized TIS model with non-integer non-negative n<1. This manuscript is well organized and shows the characteristics of the generalized n-CSTR model with various cases. However, spell checks are required for the entire contents of the manuscript (ex. line 38: steam -> stream, line 44: rector -> reactor).

Author Response

Answer to reviewer 3

We thank the reviewer for the comments and for pointing out these typos. We made minor language revisions throughout the manuscript.